# MiCoTA: Bridging the Learnability Gap with Intermediate CoT and Teacher Assistants

## Abstract

Large language models (LLMs) excel at reasoning tasks requiring long thought sequences for planning, reflection, and refinement. However, their substantial model size and high computational demands are impractical for widespread deployment. Yet, small language models (SLMs) often struggle to learn long-form CoT reasoning due to their limited capacity, a phenomenon we refer to as the "SLMs Learnability Gap". To address this, we introduce **Mi**d-**Co**T **T**eacher **A**ssistant Distillation (MiCoTA), a framework for improving long CoT distillation for SLMs. MiCoTA employs intermediate-sized models as teacher assistants and utilizes intermediate-length CoT sequences to bridge both the capacity and reasoning length gaps. Our experiments on downstream tasks demonstrate that although SLMs distilled from large teachers can perform poorly, by applying MiCoTA, they achieve significant improvements in reasoning performance. Specifically, Qwen2.5-7B-Instruct and Qwen2.5-3B-Instruct achieve an improvement of 3.47 and 3.93 respectively on average score on AIME2024, AMC, Olympiad, MATH-500 and GSM8K benchmarks. To better understand the mechanism behind MiCoTA, we perform a quantitative experiment demonstrating that our method produces data more closely aligned with base SLM distributions. Our insights pave the way for future research into long-CoT data distillation for SLMs.

## 1 Introduction

Reasoning has long been regarded as one of the most challenging capabilities to instill in Large Language Models (LLMs). The advent of Chain-of-Thought (CoT) prompting (Wei et al., 2022) marked a significant milestone, revealing the emergent reasoning abilities of large-scale models when prompted to generate step-by-step thought processes. Building on this, a lot of work (Hao et al., 2023; Yao et al., 2023; Zelikman et al., 2022; Qi et al., 2024; Wan et al., 2024) has been developed to improve LLM reasoning by scaling inference time compute. The OpenAI o1 series (OpenAI, 2024a;b) became the breaking point where they enhanced effective test-time computation by encouraging LLMs to explore possible solutions and generating longer thinking sequences during inference. Following o1, other proprietary LLMs (DeepSeek-AI et al., 2025; Seed et al., 2025) also demonstrate remarkable performance on tasks requiring intricate reasoning and decision-making. However, the substantial model size and high computational demands of these state-of-the-art reasoners make them impractical for widespread deployment, particularly in resource-constrained environments. This necessitates the development of smaller, more efficient models that retain strong reasoning capability.

Knowledge distillation (Hinton et al., 2015) is one of the most promising strategy for transferring the capabilities of large teacher models to smaller student models. In the context of long CoT, most works have focused on distilling CoT trajectories, collecting vast datasets of complex queries paired with detailed reasoning steps generated by powerful LLMs (Team, 2025a; Face, 2025; Ye et al., 2025). The primary goal is to equip small language models (SLMs) with the ability to perform long-form reasoning, thereby bridging the performance gap between teacher models and student models. Several successful attempts (DeepSeek-AI et al., 2025; Team, 2024b; Ye et al., 2025) achieved promising results at a 32 billion parameters scope.

Despite progress, SLMs still struggle to learn long-CoT reasoning effectively (Li et al., 2025; Yin et al., 2025), which could be attributed to their limited capacity. We refer to this phenomenon as "SLMs Learnability Gap" where the performance of SLMs degrades when trained on long CoT

sequences distilled from large teacher models. In this paper, we conduct an in-depth investigation to unveil and better understand the underlying causes of this phenomenon. We propose a multifaceted approach called **Mi**d-**C**o**T T**eacher **A**ssistant Distillation (MɪCoTA), illustrated in Figure 1. MɪCoTA leverages an intermediate-sized model as a teacher assistant (TA) model and distills intermediate-lengthed CoT data to train the student model.

To begin with, we conducted pivotal studies to unveil this counterintuitive phenomenon. Our experiments with models trained on distillation data from teachers of varying sizes revealed that not only did models trained with larger teachers perform worse, but even those distilled from intermediate-sized teachers could not match the performance of the original base models (Yang et al., 2024). This finding led us to explore the complementary dimension of the problem: rather than focusing solely on model size, we investigated the impact of generating intermediate-length CoT reasoning data to bridge the gap between the comprehensive reasoning of large teachers and the learning capacity of smaller students. Specifically, inspired by previous work on long-to-short reasoning (Wu et al., 2025), we found that simply merging the long-CoT data trained model with its base model resulted in a new model that produced reasoning sequences approximately half the length of the original long-CoT output without a performance drop. We used these intermediate-length CoT data from the merged TA model to train the student model. This method addresses the SLMs Learnability Gap by mitigating both the capacity gap between the student and teacher models, as well as the length gap by learning from intermediate-length CoT, allowing smaller models to better benefit from large reasoning LLMs.

We apply our method to Qwen-series SLMs (Yang et al., 2024) spanning 1.5B, 3B and 7B parameters and evaluate them across various benchmarks (Hendrycks et al., 2021; He et al., 2024; Cobbe et al., 2021). Our experiments demonstrate that MɪCoTA has significantly improved the reasoning performance of SLMs comparing to those directly trained on distilled data from the teacher model. Specifically, applying MɪCoTA results in a 9.98% improvement in reasoning performance for Qwen2.5-3B-Instruct over its base model and a 35.6% increase over its version distilled from a larger teacher model. We also conduct ablation studies to thoroughly analyze the effectiveness of our method.

Our main contributions are summarized as follows:

1. We propose a novel approach, Mid-CoT Teacher Assistant Distillation (MɪCoTA), which effectively mitigates the SLMs Learnability Gap and achieves significant improvements in reasoning performance for SLMs.

2. We conduct thorough extensive studies to analyze the impact of various components of the MɪCoTA approach. These studies confirm the effectiveness of our method and provide insights into the factors that contribute to improved reasoning performance for smaller models.

## 2 RELATED WORK

### 2.1 KNOWLEDGE DISTILLATION FOR LLM REASONING

Knowledge distillation (KD) has long been a fundamental technique for transferring knowledge from large, powerful teacher models to student models (Hinton et al., 2015; Sanh et al., 2019; Sun et al., 2019; Jiao et al., 2020; Wang et al., 2020; Zhou et al., 2022; Xu et al., 2021). Early works in KD focused on matching the output logits of the teacher and student (Hinton et al., 2015; Beyer et al., 2022; Tang et al., 2019; Park et al., 2021; Zhao et al., 2022; Sanh et al., 2019), while subsequent research extended this idea to internal representations such as attention maps and hidden states (Romero et al., 2015; Sun et al., 2019; Zagoruyko & Komodakis, 2016; Wang et al., 2022; Jiao et al., 2020; Wang et al., 2020; Xu et al., 2020).

Beyond logits distillation (Hinton et al., 2015) and internal feature distillation (Wang et al., 2020; 2022), sequence-level distillation (Kim & Rush, 2016) has also gained traction since the rise of large language models (LLMs). Previous works (Wang et al., 2023; Xu et al., 2023; Taori et al., 2023; Chiang et al., 2023) explored distilling knowledge from GPT-4 by generating synthetic instruction data and leveraging teacher-generated responses for student training.

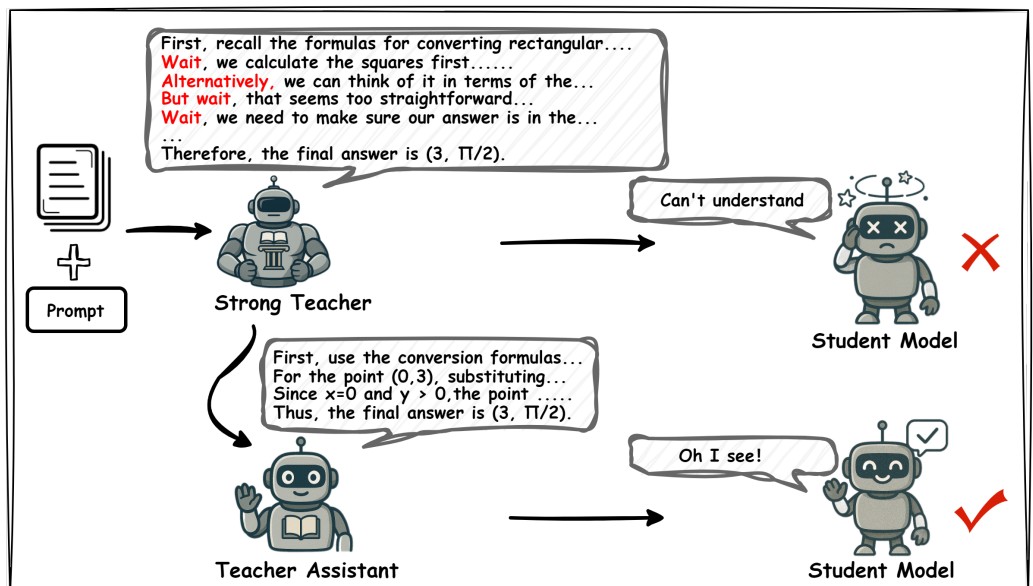

Figure 1: This figure illustrates the procedure and core concept of MICoTA. While long CoT data distilled from a strong teacher is too lengthy for the student model to effectively learn, the half-length CoT data generated by an intermediate-sized Teacher Assistant model is more accessible and easier for the student model to benefit from.

Recently, Chain-of-Thought (CoT) data distillation has emerged as a promising direction to enhance the reasoning capabilities of student models (Team, 2025a; Face, 2025; Wen et al., 2025; Li et al., 2025). By training student models on reasoning traces generated by strong LLMs, these methods aim to transfer complex multi-step reasoning skills. However, directly distilling long CoT traces from strong teachers often overwhelms smaller models, leading to suboptimal performance (Li et al., 2025).

## 2.2 THE LEARNABILITY GAP

The learnability gap—the challenge arising from large discrepancies in capacity or architecture between teacher and student models—has been recognized as a long-standing issue in knowledge distillation (Mirzadeh et al., 2020; Zhang et al., 2023b). To address this, the teacher assistant (TA) paradigm introduces an intermediate model that bridges the gap between the strong teacher and the small student (Mirzadeh et al., 2020; Son et al., 2021; Zhang et al., 2023a). TAs have been implemented across various settings, including differences in model size (Mirzadeh et al., 2020), architecture, and submodules, to facilitate more effective knowledge transfer.

Despite the success of TAs in standard KD scenarios, their application in long-CoT distillation remains underexplored. Few studies have investigated the use of TAs to address the length aspect of the gap curse in CoT distillation. Recent work (Li et al., 2025) proposed mixing long-CoT and short-CoT data to ease the learning burden on student models. On the other hand, our approach proposes directly generating intermediate-length CoT data with a merged TA model (Goddard et al., 2024), effectively filling the gap and enabling small language models to benefit from rich teacher reasoning without being overwhelmed.

## 3 METHODOLOGY

In this section, we propose to use an intermediate-sized model as a teacher assistant (TA) to bridge the gap between teacher models and student models. The key idea is to train an LLM with both model size and CoT length set to approximately half of the teacher. We pose that not only the curse of

capacity gap happened on the size of the model, but also the length of the reasoning path. Therefore this dual "half-size, half-length" design ensures that the teacher assistant is not only more accessible for the student model to learn from in terms of capacity, but also provides reasoning demonstrations that are less overwhelming in length, thereby facilitating more effective knowledge transfer.

As illustrated in Figure 1, our Teacher Assistant is first trained using long CoT generated by the Strong Teacher. Afterward, we use the model merging method to merge the Teacher Assistant before and after fine-tuning to arrive at a final, Mid-CoT Teacher Assistant. This Mid-CoT Teacher Assistant is then employed to generate CoT that the Student Models can utilize for more effective training.

## 3.1 REASONING DATA GENERATION

We employ **R1-Distill-Qwen-32B** (DeepSeek-AI et al., 2025) as our strong teacher to generate exemplar long CoT. We craft prompts that explicitly request comprehensive reasoning steps. This ensures the generated CoT captures intermediate reasoning processes rather than just final answers. For each prompt, the strong teacher produces a detailed long CoT trace, including intermediate reasoning steps and the final answer. After data generation, we further exclude any entries with incorrect answers or those that exceed the maximum length to ensure the quality of the dataset. We compile these traces into a dataset:

$$D_{strong} = \left\{ \left( x^{(i)}, CoT_{strong}^{(i)} \right) \mid A_i = \text{true and } L_i \leq L_{\max} \right\}_{i=1}^{N}, \tag{1}$$

where $A_i$ indicates whether the answer is correct, $L_i$ represents the length of the generated CoT, $L_{\max}$ is the maximum allowable length, $x^{(i)}$ is the prompt, and $CoT_{strong}^{(i)}$ is the corresponding long CoT.

## 3.2 TEACHER ASSISTANT CREATION

**Training**  We select a medium-scale model with parameters less than those of the Strong Teacher but greater than those of the Student Model. We then fine-tune this model on $D_{strong}$, resulting in a model that learns to approximate the reasoning style of the Strong Teacher.

**Model Merging**  Previous research (Wu et al., 2025) has shown that model merging effectively facilitates long-to-short reasoning by combining the quick-thinking abilities of System 1 models with the analytical reasoning of System 2 models. This approach directly manipulates model parameters without additional training. Figure 2 illustrates that model merging can reduce the response lengths of System 2 models by approximately half while maintaining performance.

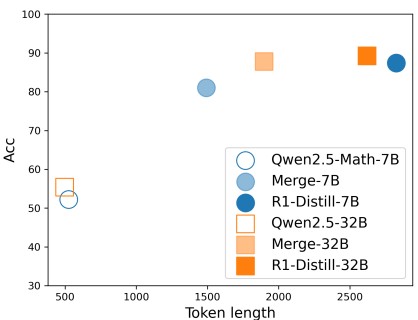

Figure 2: The token length is reduced to about half of the System 2 models by model merging.

To further reduce response length while balancing performance, we merge the pre-fine-tuned and post-fine-tuned versions of the Teacher Assistant model to achieve an intermediate response length. We utilize the *Dare* (Yu et al., 2024) algorithm to merge the Teacher Assistant before and after fine-tuning. *Dare* reduces interference between different models through sparsification of task-specific delta vectors. It utilizes random pruning in conjunction with a rescaling technique to preserve the original models' performance (Goddard et al., 2024).[1] In our experiments, we combine *Dare* pruning with the TIES sign consensus mechanism to optimize the model merging process. After model merging, we obtain Mid-CoT Teacher Assistant, an intermediate state in both model size and inference length that inherits the complementary strengths of both models without additional supervised fine-tuning.

---

[1]Refer to `https://github.com/arcee-ai/mergekit` for the model merging codebase.

Table 1: Performance gap ($\Delta$) between models trained on distillation CoT data and the original base models.

| Student Models | QwQ-32B | Avg | $\Delta$ | R1-Distill-Qwen-32B | Avg | $\Delta$ |
|---|---|---|---|---|---|---|
| 14B-Instruct | +Teacher | 55.84 | +3.66 | +Teacher | 54.90 | +2.10 |
| 1.5B-Instruct | +Teacher | 23.29 | −6.97 | +Teacher | 21.25 | −9.01 |
| | +TA | 25.16 | −5.10 | +TA | 24.59 | −5.67 |
| 3B-Instruct | +Teacher | 32.00 | −7.36 | +Teacher | 31.92 | −7.44 |
| | +TA | 34.63 | −4.73 | +TA | 33.08 | −6.28 |

## 3.3 STUDENT MODEL TRAINING

We prompt the Mid-CoT Teacher Assistant to produce a synthetic medium-length CoT dataset $D_{MiCoTA} = \left\{ (x^{(i)}, CoT_{MiCoTA}^{(i)}) \right\}_{i=1}^{N}$. Unlike the original long CoT from the Strong Teacher, these new CoT sequences are shorter and presumably more accessible for the Student Models to learn. Using $D_{MiCoTA}$, we fine-tune the Student Models with standard supervised fine-tuning (SFT).

## 4 EXPERIMENTS

In this section, we give a detailed introduction to our pivotal study, experimental setup, main results, ablation analysis, etc.

## 4.1 PIVOTAL STUDY

This section evaluates the performance of student models trained on Strong Teacher CoT and Intermediate-sized Teacher CoT. We choose QwQ-32B (Team, 2025b) and R1-Distill-Qwen-32B (DeepSeek-AI et al., 2025) as our Strong Teacher models to generate Teacher CoT responses. The Qwen2.5-14B-Instruct model, trained on CoT data generated by Strong Teachers, serves as the Intermediate-sized Teacher to produce TA CoT. Student models include three parameter-scale models: Qwen2.5-1.5B-Instruct, Qwen2.5-3B-Instruct, and Qwen2.5-14B-Instruct. To quantify the impact of different distillation data on student models, we define a performance difference metric $\Delta$ to measure the performance gap between trained models and their original baseline models, defined as:

$$\Delta = P_{distilled} - P_{base}. \tag{2}$$

Table 1 provides the results. A negative (positive) $\Delta$ indicates that the performance of models trained on CoT data is worse (better) than that of the original base models. The results indicate a nuanced relationship between student model size and performance when trained on distillation data from the Strong Teacher. Specifically, the larger student model, Qwen2.5-14B-Instruct, demonstrates a clear advantage when trained using the Strong Teacher CoT, achieving a positive performance gain. Conversely, smaller models (1.5B-Instruct and 3B-Instruct) experience negative performance changes when trained on the same CoT data, suggesting they may struggle to effectively learn from the longer CoT paradigm. Subsequently, we utilize the well-trained Qwen2.5-14B-Instruct as our Intermediate-sized Teacher Assistant model to generate CoT data for training the smaller student models. The results show that the Intermediate-Sized Teacher Assistant data provides improvements for the smaller models, but still cannot match the performance of the original base model.

## 4.2 EXPERIMENTAL SETUP

**Datasets** We utilize a prompt set of 12K items from the MATH dataset (Lightman et al., 2023), which includes seven math subjects: advanced calculus, geometry, and linear algebra. For data generation, we apply greedy decoding with a maximum token limit of 16K. During this process, we exclude any entries that exceed the maximum length. By pairing math problem instructions with corresponding solutions from teacher models, we create problem-solution pairs for fine-tuning the student models.

**Models**  We employ R1-Distill-Qwen-32B (DeepSeek-AI et al., 2025) as our strong teacher model. We utilize the Qwen2.5-14B-Instruct as our teacher assistant model. For the student models, we evaluate three SLMs from the Qwen family, which cover a range of parameter scales: Qwen2.5-7B-Instruct, Qwen2.5-3B-Instruct, and Qwen2.5-1.5B-Instruct.

**Training**  All fine-tuning experiments were performed using the LLaMA-Factory framework (Zheng et al., 2024) on a server equipped with eight NVIDIA A800-SXM4-80GB GPUs. Detailed hyperparameters and information about the training setting are provided in Appendix A.

**Evaluation**  We evaluate performance on the following benchmarks: AIME 2024, AMC 2023, OlympiadBench (He et al., 2024), MATH 500 (Hendrycks et al., 2021), and GSM8K (Cobbe et al., 2021). To make our empirical results more reproducible, we adopt greedy decoding during inference. All models are assessed in a zero-shot setting. The maximum length for the evaluated models is set to 16K tokens. Following Gao et al. (2024), we only adopt a rule-based method to extract the final answer out of model's generation and measure the exact-match accuracy[2]. We do not apply LLM-as-judge based evaluation in our experiments.

**Baselines**  We compare MICOTA against three baseline settings:

- **Instruct**: SLMs without trained on long-CoT data.
- **Strong Teacher CoT**: The SLMs trained on Long-CoT data generated by the Strong Teacher.
- **Mix-Long** (Li et al., 2025): Fine-tuning on data mixed with long CoT and short CoT data, where long CoT is generated from QwQ-32B and short CoT is generated from Qwen2.5-32B-Instruct.

**Model Setting**

- **Half-size CoT**: The SLMs trained on long-CoT data distilled from Intermediate-sized Teacher Assistant.
- **MICOTA**: The SLMs trained on intermediate-length CoT data distilled from merged Teacher Assistant.

## 4.3 MAIN RESULT

Table 2 presents a comprehensive evaluation of model performance across different benchmarks. From the table, we can observe a performance decline in the student models after training with the Strong Teacher CoT. This finding is consistent with previous studies (Li et al., 2025; Wu et al., 2025), which suggest that smaller models often struggle to effectively learn from stronger teacher models. For instance, after training on the Strong Teacher CoT, the Qwen2.5-7B model exhibits a decrease in its average score compared to the instruct models without any training. When trained with the Mix-Long method, the model's performance improves. In contrast, the student models trained using the Mid-CoT Teacher Assistant outputs exhibit improvements. For example, the Qwen2.5-7B model achieves an average score of 49.36, surpassing all baseline methods. Similarly, the Qwen2.5-3B student model shows enhanced performance, with an average score of 43.29 when trained with the MICOTA. The Qwen2.5-1.5B model also benefits from our approach, with its average score increasing to 33.01. We also extend our experiments to LLaMA family models in Table 8 and provide detailed LLM-judged benchmark scores in Table 9 (Appendix B). These findings collectively demonstrate that our method, which leverages the expertise of the Teacher Assistant for fine-tuning student models, effectively enhances their overall performance.

## 4.4 ABLATION STUDIES

To validate the impact of our design choices, we conducted ablation studies on Qwen-Instruct models of varying sizes. Table 3 presents the results of different distillation methods, which include Strong

---

[2]Refer to `https://github.com/EleutherAI/lm-evaluation-harness` for the evaluation codebase.

Table 2: Performance comparison across various benchmarks; the highest score is bolded, and the second highest score is underlined.

| Models | Method | AIME | AMC | Olympiad | MATH-500 | GSM8K | Average |
|---|---|---|---|---|---|---|---|
| **Strong Teacher** | | | | | | | |
| R1-Distill-Qwen-32B | – | 60.00 | 90.00 | 41.92 | 86.00 | 86.27 | 72.83 |
| **(Merged) Teacher Assistant** | | | | | | | |
| Qwen2.5-14B | – | 23.33 | 62.50 | 31.55 | 77.40 | 87.26 | 56.40 |
| **Student Models** | | | | | | | |
| | Instruct | 10.00 | 40.00 | 24.44 | **72.60** | 82.41 | 45.89 |
| Qwen2.5-7B | Strong Teacher CoT | 6.66 | 27.50 | 20.00 | 64.20 | 84.38 | 40.54 |
| | Mix-Long* | 10.00 | 37.50 | 24.44 | 68.40 | **88.85** | 45.83 |
| | **MICOTA** | **13.33** | **52.50** | **25.62** | 70.40 | 84.98 | **49.36** |
| | Instruct | 3.33 | 35.00 | 18.07 | **62.80** | 77.63 | 39.36 |
| Qwen2.5-3B | Strong Teacher CoT | 3.33 | 22.50 | 12.59 | 46.20 | 74.98 | 31.92 |
| | Mix-Long* | 10.00 | 37.50 | 19.11 | 60.20 | **81.50** | 41.66 |
| | **MICOTA** | **13.33** | **40.00** | **20.59** | 61.60 | 80.97 | **43.29** |
| | Instruct | **3.33** | 22.50 | 12.44 | 45.20 | 67.85 | 30.26 |
| Qwen2.5-1.5B | Strong Teacher CoT | 0.00 | 7.50 | 7.85 | 33.00 | 57.92 | 21.25 |
| | Mix-Long* | **3.33** | 25.00 | 11.70 | 50.80 | **71.65** | 32.49 |
| | **MICOTA** | **3.33** | **27.50** | **13.03** | **51.00** | 70.02 | **33.01** |

\* The results presented here are based on our reproduction of the method.

Table 3: Ablation studies on MICOTA with Qwen2.5-7B-Instruct, Qwen2.5-3B-Instruct, and Qwen2.5-1.5B-Instruct. The highest score is bolded, and the second highest score is underlined.

| Models | Distillation Method | AIME | AMC | Olympiad | MATH-500 | GSM8K | Average |
|---|---|---|---|---|---|---|---|
| | Strong Teacher CoT | 6.66 | 27.50 | 20.00 | 64.20 | 84.38 | 40.54 |
| Qwen2.5-7B | Half-size CoT | 10.00 | 42.50 | 19.70 | 66.20 | **86.27** | 44.93 |
| | **MICOTA** | **13.33** | **52.50** | **25.62** | **70.40** | 84.98 | **49.36** |
| | Strong Teacher CoT | 3.33 | 22.50 | 12.59 | 46.20 | 74.98 | 31.92 |
| Qwen2.5-3B | Half-size CoT | 3.33 | 22.50 | 12.30 | 50.40 | 76.88 | 33.08 |
| | **MICOTA** | **13.33** | **40.00** | **20.59** | **61.60** | **80.97** | **43.29** |
| | Strong Teacher CoT | 0.00 | 7.50 | 7.85 | 33.00 | 57.92 | 21.25 |
| Qwen2.5-1.5B | Half-size CoT | 0.00 | 22.50 | 7.55 | 33.80 | 59.13 | 24.59 |
| | **MICOTA** | **3.33** | **27.50** | **13.03** | **51.00** | **70.20** | **33.01** |

Teacher CoT, Half-size CoT, and MICOTA. These denote models trained on CoT data generated by the Strong Teacher, CoT data produced by the Intermediate-sized Teacher, and CoT data generated by our Mid-CoT Teacher Assistant, respectively. The results illustrated in Table 3 indicate that the student models trained on Half-size CoT outperform those trained on Strong Teacher CoT, highlighting its effectiveness as an intermediate-sized teacher model. Implementing an intermediate-sized teacher for small language models (SLMs) provides adequate guidance without overwhelming them. Notably, our MICOTA approach shows a clear upward trend in performance across all model sizes, consistently surpassing the previous methods and achieving the highest scores among all configurations tested. Our MICOTA strikes a moderate balance in both model scale and response length: it preserves sufficient reasoning depth without causing information overload, thereby enabling SLMs to internalize complex multi-step reasoning patterns more effectively. We provide a case study

of Half-size CoT and MICoTA output in Appendix C. These results confirm the effectiveness of our proposed MICoTA approach.

To verify the sensitivity of the results to the size of the Teacher Assistant (TA), we merge DeepSeek-R1-Distill-Qwen-32B and Qwen2.5-32B-Instruct as the half-length TA to train Qwen2.5-7B, Qwen2.5-3B, and Qwen2.5-1.5B student models. The model-merge strategy and all student-model training hyper-parameters are identical to those used when Qwen2.5-14B-Instruct served as the half-length teacher assistant. As shown in Table 4, Merged 32B teacher (half-length CoT) yields best performance for 7B student. For smaller models (3B/1.5B), 14B TA proves more effective. These results strengthen our original claims and provide more comprehensive evidence for MICoTA's effectiveness and general applicability. The detailed results regarding sensitivity to the merge method and merge ratio are provided in Table 10 and Table 11 of Appendix B.

Table 4: Performance of Student Models Trained with Merged 32B Models as Half-Length Teacher Assistants.

|  | AIME | AMC | Olympiad | MATH-500 | GSM8K | Average | Merged 14B as TA avg |
|---|---|---|---|---|---|---|---|
| Qwen2.5-7B | 16.66 | 50.00 | 25.18 | 73.00 | 86.20 | 50.21 | 49.36 (-0.85) |
| Qwen2.5-3B | 3.33 | 35.00 | 19.11 | 61.40 | 81.57 | 40.10 | 43.29 (+2.19) |
| Qwen2.5-1.5B | 6.66 | 27.50 | 13.33 | 47.20 | 68.15 | 32.57 | 33.01 (+0.44) |

Table 5: The results of adaptability of models on different CoT data.

| Method \ Model | Qwen2.5-32B | Qwen2.5-14B | Qwen2.5-7B | Qwen2.5-3B | Qwen2.5-1.5B |
|---|---|---|---|---|---|
| Strong Teacher CoT | 0.17 | 0.18 | 0.20 | 0.26 | 0.21 |
| Half-size CoT | 0.17 | 0.17 | 0.19 | 0.24 | 0.20 |
| Mix-Long | 0.13 | 0.14 | 0.14 | 0.15 | 0.16 |
| **MICoTA** | **0.11** | **0.11** | **0.13** | **0.13** | **0.14** |

## 4.5 ADAPTABILITY OF MODELS ON DIFFERENT COT DATA

We adopt the adapted Bits Per Character (BPC) metric from (Zhu et al., 2025), which is a variation of perplexity that eliminates length differences, to evaluate the model's adaptability with the text. The calculation is as follows:

$$BPC(T) = \frac{-\sum_{i=1}^{N} \log p(w_i | w_1, \cdots, w_{i-1})}{\text{len-utf-8}(T)} \quad (3)$$

Here, $T$ denotes the text under analysis, $N$ is the number of tokens in $T$, and len-utf-8$(T)$ indicates the length of $T$ in UTF-8 encoding, measured in characters. The tokens $w_i$ are segments of text. By employing this metric, we are able to assess the models' orientations toward various distributions of CoT data and identify potential gaps in their learning processes. For this evaluation, we selected the models Qwen2.5-32B-Instruct, Qwen2.5-14B-Instruct, Qwen2.5-7B-Instruct, Qwen2.5-3B-Instruct, and Qwen2.5-1.5B-Instruct to analyze their adaptability on the Strong Teacher CoT data, Half-size CoT data, Mix-Long data, and the proposed MICoTA data. The results are summarized in Table 5, which illustrates the BPC values across different models and data types. For the Strong Teacher CoT data, the smaller models exhibit relatively high BPC values, reflecting a larger distribution gap between the model predictions and the actual data distribution. This discrepancy may contribute to their difficulties in effectively learning from the Strong Teacher CoT data. In contrast, the BPC values for the Half-size CoT data are slightly lower than those for the Strong Teacher CoT data, suggesting a reduced distribution gap. This reduction indicates that the smaller models can adapt slightly better from the intermediate-sized teacher. Notably, for the MICoTA data, we observe considerably lower

BPC values across all models compared to the Strong Teacher CoT, Half-size CoT, and Mix-Long data. This could suggest that the MICOTA CoT data more closely matches their inherent distribution, thereby facilitating a more effective learning process. These findings validate the effectiveness of our proposed method.

## 4.6 LENGTH ANALYSIS

We further explore response lengths of student models under three configurations: the Instruct model, the one trained with Strong Teacher CoT data, and the one trained with MICOTA CoT data. As shown in Figure 3, response lengths differ significantly across models and configurations. Across all sizes, the Instruct model—designed for concise, targeted answers—produces shorter responses. By contrast, the Strong Teacher CoT-trained version, which emphasizes chain-of-thought (CoT) reasoning, generates longer outputs via step-by-step analysis and elaboration. The version trained with MICOTA, meanwhile, finds a balance between brevity and comprehensiveness, yielding outputs of medium length as it incorporates some elements of chain-of-thought reasoning without excessive elaboration. Additionally, response lengths generally increase as model size decreases. Smaller models, with limited parameter capacity, may need more verbose explanations to cover all reasoning steps; larger models integrate information more efficiently, delivering comprehensive yet succinct responses, unlike smaller ones that often rely on detailed descriptions for coherence and completeness. We further analyze **top 30** high-entropy tokens (Wang et al., 2025) in the reasoning responses of **DeepSeek-R1-Distill-Qwen-32B** and **MICOTA** in Table 6. These tokens are considered strongly associated with successful model reasoning. The relative ranking of word choices reveals distinct reasoning styles: 1) **MICOTA** prioritizes concise, imperative terms (e.g., "try," "consider") and formal logic markers (e.g., "by," "given"), reflecting a **structured, theorem-like** approach. 2) **DeepSeek** exhibits self-correction (e.g., "wait," "perhaps," "actually") and speculative language (e.g., "maybe," "think"), suggesting **exploratory, iterative** reasoning.

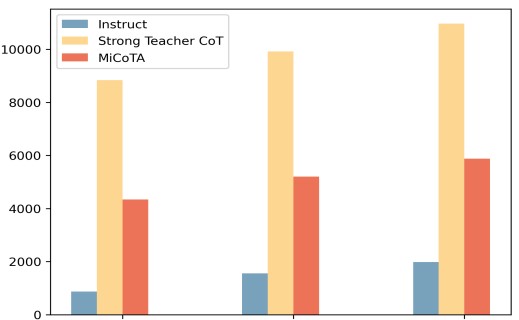

| High Entropy Token | MICOTA Rank | DeepSeek Rank |
|---|---|---|
| try | **22** | — |
| consider | **24** | — |
| by | **16** | — |
| given | **27** | — |
| from | 9 | **15** |
| wait | **13** | 4 |
| perhaps | — | **29** |
| actually | — | **30** |
| maybe | **17** | 9 |
| think | **23** | 19 |

Table 6: High Entropy Token Ranking

Figure 3: Comparison of Response Lengths (in tokens) for Different Methods across Student Models.

## 5 CONCLUSION

In this paper, we propose Mid-CoT Teacher Assistant Distillation (MICOTA), a novel approach designed to address the "SLMs Learnability Gap" problem by mitigating both the capacity gap between the student and teacher models, as well as the length gap by learning from intermediate-length CoT, allowing smaller models to better benefit from large reasoning LLMs. Our extensive experiments across multiple benchmarks demonstrate that MICOTA significantly improves the reasoning performance of SLMs, achieving up to 35.6% better results compared to models directly trained on long CoT data from large teachers.

By providing thorough extensive experiments, MICOTA paves the way for more effective training of smaller models while retaining strong reasoning capabilities. Our findings highlight the potential of intermediate-length CoT sequences in fostering better reasoning outcomes in resource-constrained environments, offering a promising direction for future research in sequence-level knowledge distillation and model efficiency.

## 6 REPRODUCIBILITY STATEMENT

We ensure reproducibility: detailed information on experimental setups, hyperparameters, data preprocessing steps, evaluation metrics, and model architectures—all necessary to reproduce the main results—is provided in Section 3, Section 4, and Appendix A.

## 7 ETHICS STATEMENT

This research adheres to ICLR's Code of Ethics. We ensure that all aspects of our research, including data collection, data usage, and experimentation, adhere to these standards.

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

## A  TRAINING DETAILS

Table 7: List of all models used in our experiments, with Hugging Face links where available.

| Model Name | Hugging Face Link |
|---|---|
| QwQ-32B (Team, 2025b) | `https://huggingface.co/Qwen/QwQ-32B` |
| R1-Distill-Qwen-32B (DeepSeek-AI et al., 2025) | `https://huggingface.co/deepseek-ai/DeepSeek-R1-Distill-Qwen-32B` |
| Qwen2.5-32B-Instruct (Team, 2024a) | `https://huggingface.co/Qwen/Qwen2.5-32B-Instruct` |
| Qwen2.5-14B-Instruct | `https://huggingface.co/Qwen/Qwen2.5-14B-Instruct` |
| Qwen2.5-7B-Instruct | `https://huggingface.co/Qwen/Qwen2.5-7B-Instruct` |
| Qwen2.5-3B-Instruct | `https://huggingface.co/Qwen/Qwen2.5-3B-Instruct` |
| Qwen2.5-1.5B-Instruct | `https://huggingface.co/Qwen/Qwen2.5-1.5B-Instruct` |

### A.1  MODELS

Table 7 provides a detailed summary of the models utilized in our study.

### A.2  PARAMETERS SETTING

**Qwen2.5-14B-Instruct**   The model is trained using eight NVIDIA A800-SXM4-80GB GPUs. We set the batch size to 32 and the peak learning rate to 1e-5, following a cosine decay schedule. A weight decay of 0.01 is applied, and for all experiments, we train for a maximum of two epochs.

**Qwen2.5-7B-Instruct**   The model is trained using eight NVIDIA A800-SXM4-80GB GPUs. For all experiments, we set the batch size to 32 and the peak learning rate to 1e-5, following a cosine decay schedule. A weight decay of 0.01 and a maximum of two epochs are applied.

**Qwen2.5-3B-Instruct**   The model is trained using eight NVIDIA A800-SXM4-80GB GPUs. For the QwQ-32B strong teacher experiment, we set the batch size to 32 and the peak learning rate to 1e-5, following a cosine decay schedule. A weight decay of 0.01 and a maximum of two epochs are applied. For the R1-Distill-Qwen-32B strong teacher experiment, we set the batch size to 128 and the peak learning rate to 2e-5, also following a cosine decay schedule. A weight decay of 0.01 and a maximum of three epochs are applied.

**Qwen2.5-1.5B-Instruct**   The model is trained using eight NVIDIA A800-SXM4-80GB GPUs. For the QwQ-32B strong teacher experiment, we set the batch size to 32 and the peak learning rate to 1e-5, following a cosine decay schedule. A weight decay of 0.01 and a maximum of two epochs are applied. For the R1-Distill-Qwen-32B strong teacher experiment, we set the batch size to 96 and the peak learning rate to 2e-5, also following a cosine decay schedule. A weight decay of 0.01 and a maximum of three epochs are applied.

# B   MORE EXPERIMENTS RESULTS

## B.1   GENERALIZATION ACROSS ARCHITECTURES

We extend our experiments to models of the LLaMA family to demonstrate the generality of our approach to other architectures. Specifically, we use DeepSeek-R1-Distill-Llama-70B as the strong teacher, LLaMA-3.1-8B-Instruct as the teacher assistant, and LLaMA-3.2-3B-Instruct as the student model. As shown in Table 8, "Instruct" denotes the untrained checkpoint; "Strong teacher CoT" denotes the version trained on CoT data distilled from DeepSeek-R1-Distill-Llama-70B; "Merge" denotes the teacher assistant obtained by merging LLaMA-3.1-8B-Instruct with the checkpoint trained by the strong teacher CoT; and "MICOTA " denotes the student model trained on data distilled from the Merge teacher assistant. As demonstrated in Table 8, the advantages of MICOTA transfer effectively to the LLaMA family. The relative improvements show similar patterns to Qwen family, which further verifies the effectiveness of our proposed method.

Table 8: Ablation studies on MICOTA with LLaMA-3.1-8B, LLaMA-3.2-3B. The highest score is bolded, and the second highest score is underlined.

| Models | Method | AIME | AMC | Olympiad | MATH-500 | GSM8K | Average |
|---|---|---|---|---|---|---|---|
| LLaMA-3.1-8B | Instruct | **3.33** | **35.00** | 12.00 | 47.60 | 76.80 | 34.94 |
| | Strong Teacher CoT | **3.33** | 25.00 | 13.18 | 48.40 | **81.12** | 34.20 |
| | **MICOTA** | 0 | 30.00 | **14.66** | **55.60** | 75.73 | **35.19** |
| LLaMA-3.2-3B | Insturct | **3.33** | 22.50 | 10.07 | **46.40** | 69.74 | 30.40 |
| | Strong Teacher CoT | **3.33** | 17.50 | 10.51 | 44.00 | 73.00 | 29.66 |
| | **MICOTA** | **3.33** | **27.50** | **10.81** | 41.20 | **76.04** | **31.77** |

Table 9: Qwen-2.5-32B as Judge results; the highest score is bolded, and the second highest is underlined.

| Models | Method | AIME | AMC | Olympiad | MATH-500 | GSM8K | Average |
|---|---|---|---|---|---|---|---|
| Qwen2.5-7B | Instruct | 10.00 | 40.00 | **39.85** | **77.20** | **91.96** | 51.80 |
| | Strong Teacher CoT | 6.66 | 27.50 | 29.48 | 67.40 | 90.14 | 44.24 |
| | Mix-Long* | 10.00 | 37.50 | 38.37 | 72.00 | 89.38 | 49.45 |
| | **MICOTA** | **13.33** | **52.50** | 38.07 | 74.20 | 90.14 | **53.65** |
| Qwen2.5-3B | Instruct | 3.33 | 35.00 | 27.11 | **65.60** | 84.91 | 43.19 |
| | Strong Teacher CoT | 3.33 | 22.50 | 16.44 | 47.80 | 81.60 | 34.33 |
| | Mix-Long* | 10.00 | 37.50 | 28.74 | 63.40 | 82.71 | 44.47 |
| | **MICOTA** | **13.33** | **40.00** | **29.03** | 64.20 | **85.06** | **46.32** |
| Qwen2.5-1.5B | Instruct | **3.33** | 22.50 | **20.59** | 46.80 | **73.16** | 33.28 |
| | Strong Teacher CoT | 0.00 | 7.50 | 10.81 | 34.20 | 65.65 | 23.63 |
| | Mix-Long* | **3.33** | 25.00 | 18.07 | **54.4** | 73.08 | 34.78 |
| | **MICOTA** | **3.33** | **27.50** | 19.55 | 53.2 | 72.47 | **35.21** |

* The results presented here are based on our reproduction of the method.

## B.2 LLM AS JUDGE

Table 9 shows the detailed performance scores of each benchmark for different student models, with Qwen-2.5-32B serving as the judge. The results in the table indicate that our method effectively enhances their overall performance.

## B.3 MERGE METHOD ABLATION

To further investigate the impact of merging strategies on performance, we conduct a merge method ablation study. We merge Qwen2.5-14B-Instruct with its Strong-teacher-CoT-trained counterpart using merge ratios of 0.5 and 0.5 (for the DARE method, we adopt 0.5 as target model density) to obtain 3 variants of the teacher assistant (TA), which are then used to distill the student model. As shown in Table 10, the performance is sensitive to the merging method selection. Specifically, the linear merging method attains the best performance for the 7B student model, with a score of 51.70. On the other hand, for the 3B and 1.5B student models, the DARE method outperforms both the linear and SLERP methods. Therefore, we adopt the DARE method in our paper to enhance the performance of small language models.

Table 10: Performance of Different Merge Methods across Model Sizes

| Size | Linear | SLERP | DARE |
| --- | --- | --- | --- |
| Qwen2.5-7B | **51.70** | 49.88 | 49.36 |
| Qwen2.5-3B | 41.25 | 40.17 | **43.29** |
| Qwen2.5-1.5B | 31.56 | 31.90 | **33.01** |

## B.4 EFFECT OF MERGE RATIO

We adopt linear merging as the model merging strategy to explore the effect of merge ratio. We merge Qwen2.5-14B-Instruct with its Strong-teacher-CoT-trained counterpart using merge ratios of (0.9, 0.1), (0.1, 0.9) and (0.5,0.5) to obtain 3 variants of the teacher assistant (TA), which are then used to distill the student model.

As shown in Table 11, the experimental results reveal a clear pattern: a higher proportion of instruction (0.9Inst + 0.1CoT) yields better performance, especially for smaller student models. Specifically, for the 3B and 1.5B models, the (0.9Inst + 0.1CoT) configuration achieves the highest average scores (**44.08** and **33.56** respectively), outperforming both the balanced ratio (0.5:0.5) and the CoT-dominant ratio (0.1:0.9) by significant margins. Only the 7B model deviates from this trend, with the balanced ratio (0.5Inst + 0.5CoT) achieving the highest score (**52.70**), indicating that larger student models may have greater capacity to leverage both instruction and CoT knowledge synergistically.

Table 11: Average Score across Benchmarks (AIME2024, AMC, Olympiad, Math-500 and GSM8k)

| Student Size | 0.9Inst + 0.1CoT | 0.5Inst + 0.5Cot | 0.1Inst + 0.9 CoT |
| --- | --- | --- | --- |
| Qwen2.5-7B | 48.10 | **52.70** | 44.19 |
| Qwen2.5-3B | **44.08** | 41.25 | 33.636 |
| Qwen2.5-1.5B | **33.56** | 31.56 | 29.038 |

## C  CASE STUDY OF DIFFERENT OUTPUT

In this case study, we analyze the performance of the student model trained on Half-size CoT and MICOTA, respectively. Half-size CoT is generated from Intermediate-sized Teacher Assistant and MICOTA is generated from Mid-CoT Teacher Assistant. As shown in Figure 4, the models trained on MICOTA and Half-size CoT both exhibit the thinking patterns characteristic of Strong Teacher CoT, particularly the strategy of reflection. However, the model trained on Half-size CoT exhibited excessive instances of "wait," amounting to 209 occurrences throughout its response, indicating a tendency to over-analyze along with becoming bogged down in repetitive checks. In contrast, the model trained on MICOTA used "wait" only once. After providing the correct answer, it checked its solution briefly to ensure its accuracy, thereby concluding its response effectively. In summary, the model trained on MICOTA not only reached the correct conclusion but also achieved a balanced CoT length.

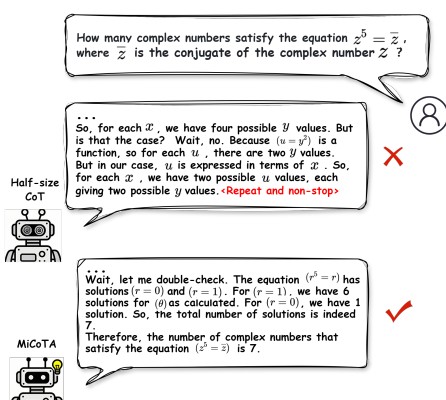

Figure 4: Case Study of MICOTA. Models trained on Half-size CoT tend to overthink, whereas those trained on MICOTA maintained a balanced reasoning path and reached the correct answer.

## D  LIMITATIONS

While our study presents improvements in addressing the small language model (SLM) learnability gap in long chain-of-thought (CoT) reasoning, our experiments are primarily conducted in the context of math reasoning tasks, and the generalizability of our findings to other domains—such as natural language understanding, code generation, or complex multi-modal tasks—remains to be explored. Future work should investigate how diverse task scenarios affect the robustness of our proposed framework, offering a more comprehensive understanding of its applicability across a broader range of applications.

## E  THE USE OF LARGE LANGUAGE MODELS (LLMS)

During the research process, Large Language Models (LLMs) offer assistance in two key areas: refining written content and offering guidance on LaTeX operations.

