# OpenReview forum: "MiCoTA: Bridging the Learnability Gap with Intermediate CoT and Teacher Assistants"
_ICLR.cc/2026/Conference — Submitted to ICLR 2026_

### Official Review · Reviewer_KAks · 2025-10-30

**Soundness:** 3
**Presentation:** 3
**Contribution:** 2
**Rating:** 4
**Confidence:** 3

**Summary:**

This paper addresses the reasoning ability gap between large and small language models in Chain-of-Thought (CoT) reasoning distillation. The authors observe that small models often fail to learn effectively from long reasoning traces generated by powerful teachers. To tackle this, they propose MICOTA (Mid-CoT Teacher Assistant Distillation), a framework that introduces an intermediate-sized “Teacher Assistant” (TA) model and leverages intermediate-length CoT sequences to bridge both the capacity gap and the reasoning-length gap between teacher and student models.

**Strengths:**

1. Clear problem definition. This paper clearly identifies the SLM learnability gap as a crucial issue in long-CoT distillation, framing it along both capacity and reasoning-length dimensions.
2. Intuitive idea. The “half-size, half-length” strategy is intuitive and well-grounded. Combining teacher-assistant distillation with model-merging (DARE + TIES) to generate mid-length CoTs is novel in the CoT context.
3. Thorough experimental validation.  The experimental results are across multiple model scales and benchmarks. Ablations confirm the method contributes to the performance.

**Weaknesses:**

1. Lack of theoretical grounding. The key novelty is their combination for “half-size, half-length” CoTs. The paper does not provide a principled criterion for how much to shorten CoTs or why the proposed merge yields half length beyond an anecdotal trend and a qualitative claim about “approximately half” tokens.
2. Evaluation is narrow on math.  All five core benchmarks are mathematical or math-heavy. This makes it unclear whether MICOTA generalizes to non-mathematical reasoning. The current evidence may only reflect math-specific artifacts rather than a general CoT length–capacity phenomenon.
3. Evaluation protocol is limited and may mask errors. The paper uses greedy decoding, a single maximum length, and exact-match via rule-based answer extraction only. Many math benchmarks require robust extraction guards to avoid format drift. The absence of calibration curves or solution-consistency checks weakens claims about genuine reasoning improvement.

**Questions:**

1. I wonder whether “half-size, half-length” CoTs really play a role. I hope more evidence shows that this method is better than usual distillation.
2. The failure analysis is not enough. Figure 1 can not prove this situation.
3. The efficiency of investing compute between MICOTA and other models?
4. As the paper only uses Qwen, what about other models? This is important for generalizability.

---

### Official Review · Reviewer_YsLS · 2025-10-31

**Soundness:** 2
**Presentation:** 2
**Contribution:** 2
**Rating:** 4
**Confidence:** 3

**Summary:**

This paper addresses the "SLMs Learnability Gap", where small language models (SLMs) struggle to learn long-form Chain-of-Thought (CoT) reasoning from large language models (LLMs) due to limited capacity. It introduces the MiCOTA framework, which uses intermediate-sized teacher assistants to generate intermediate-length CoT sequences, significantly improving SLMs' long CoT distillation performance, as demonstrated by notable gains on multiple mathematical reasoning benchmarks.

**Strengths:**

1. The framework design is reasonable and has a certain novelty: The paper addresses a practical issue in Large Language Model (LLM) knowledge distillation and proposes a clear solution with sufficient motivation. The concept of "learning capability gap" relatively accurately summarizes the problem, and the proposed "teacher-assistant-student" pipeline is a reasonable and logically consistent approach to bridge this gap.
2. The experimental results are convincing and show good performance: The main experimental validation process is detailed and has strong argumentative power. The authors demonstrate obvious performance improvements across multiple student models and a series of challenging reasoning benchmarks; moreover, the comparison with the "strong teacher CoT" baseline directly provides support for the core hypothesis.

**Weaknesses:**

1. Domain Generalization：All evaluations are conducted on mathematical reasoning tasks. It remains unclear whether MiCOTA generalizes to other reasoning-heavy domains (e.g., code reasoning, legal text analysis, multi-hop QA).
2. Faithfulness and Error Propagation in Mid-CoT：This methodology relies on a Teacher Assistant that is not perfect, leading to the risk of error propagation that has not been fully addressed. If the intermediate CoT generated by the TA has flaws, omissions, or misunderstandings, it may be systematically learned by the student model. The paper does not explore the potential quality degradation in the knowledge transfer process, which raises concerns about the reliability and robustness of the final student model.
3. Analysis of practical application feasibility is missing: The paper fails to address a key practical issue of the proposed scheme, namely, it does not resolve the significant computational overhead introduced by the MiCOTA framework. Given that efficiency is one of the main motivations for using SLMs, it is difficult to evaluate the practical application value of this method without discussing the trade-off between performance gain and additional costs.
4. Ablation on TA Size and CoT Length：The “half-size, half-length” heuristic is intuitive but untested.

**Questions:**

1. Add more intuitive examples in the appendix: For the same complex problem, present the CoT outputs of the teacher, assistant, and student models to intuitively demonstrate the step-by-step simplification of reasoning complexity. Meanwhile, provide the prompt for LLM-as-Judge to facilitate readers in reproducing the relevant evaluations.
2. The current experimental scenarios have limitations: Experiments are mainly focused on the field of mathematical reasoning. If they can further cover a wider range of tasks such as commonsense reasoning, multi-hop question answering, and scientific problem-solving, it will more fully prove the generality and practical value of the method.
3. It is suggested to supplement the analysis of boundary cases: The effectiveness of MiCOTA may be affected by the capability gap between the student and assistant models. It is recommended to systematically explore two types of boundary cases. First, when the student model is extremely small, whether the learning capability gap is too large to be bridged even with the introduction of an assistant. Second, when the capabilities of the student and assistant models are close, whether the performance improvement is close to negligible. Such analysis will help clarify the optimal application scenarios of the method.

---

### Official Review · Reviewer_kFWh · 2025-11-03

**Soundness:** 2
**Presentation:** 3
**Contribution:** 2
**Rating:** 4
**Confidence:** 2

**Summary:**

This paper addresses the challenge of small language models (SLMs) struggling to learn long-form Chain-of-Thought (CoT) reasoning when distilled from large teacher models—a phenomenon termed the "SLMs Learnability Gap." The authors propose MICOTA (Mid-CoT Teacher Assistant Distillation), a framework that employs intermediate-sized models as teacher assistants and utilizes intermediate-length CoT sequences to bridge both the capacity gap and reasoning length gap. The method involves training an intermediate-sized model (14B) on long CoT data from a strong teacher (32B), then using model merging (DARE algorithm) to create a "Mid-CoT Teacher Assistant" that generates medium-length CoT sequences. Small student models (1.5B, 3B, 7B) are then trained on this intermediate data.

**Strengths:**

- Clear problem motivation with concrete experimental evidence
- Well-structured paper with logical flow from problem identification to solution
- Addresses a practical problem in deploying reasoning-capable SLMs

**Weaknesses:**

- No rigorous theoretical explanation for why intermediate-length CoT should help beyond the intuitive capacity/length gap argument
- Only evaluated on math reasoning tasks (AIME, AMC, Olympiad, MATH-500, GSM8K)
- Missing analysis of what happens with different TA sizes

**Questions:**

- Can you provide theoretical or empirical evidence for why half-length CoT is optimal? Why not 1/3 or 2/3 length?
- How do you select the optimal TA size?
- Can you test MICOTA on other model families (LLaMA, Mistral) to show generalization?

---

### Meta-Review · Area_Chair_eVPV · 2026-01-04

**Summary:**

- A recurring concern among the reviewers is the lack of a rigorous theoretical or principled justification for why intermediate-length (“half-length”) Chain-of-Thought (CoT) sequences and intermediate-sized teacher assistants constitute an optimal design choice.

- Reviewers note that the exploration of design choices and boundary conditions is limited, particularly due to the absence of systematic ablation studies on teacher assistant (TA) size and CoT length beyond the selected “half-size, half-length” setting.

- All reviewers emphasize that the experimental evaluation is restricted to mathematical reasoning benchmarks (AIME, AMC, Olympiad, MATH-500, GSM8K), raising concerns about the generality of the empirical findings.

**Reviewer Concerns:**

No response from the authors

**Reviewer Scores:**

No response from the authors. I expect all reviewers to maintain their original score

---

### Decision · Program_Chairs · 2026-01-26

Reject